# Localization Strategy Prior to Radiofrequency Ablation for Primary and Secondary Hyperparathyroidism

**DOI:** 10.3390/biomedicines11030672

**Published:** 2023-02-23

**Authors:** Chih-Ying Lee, Yen-Hsiang Chang, Pi-Ling Chiang, Cheng-Kang Wang, An-Ni Lin, Chi-Cheng Chen, Yi-Fan Chen, Shun-Yu Chi, Fong-Fu Chou, Wei-Che Lin

**Affiliations:** 1Department of Diagnostic Radiology, Kaohsiung Chang Gung Memorial Hospital and Chang Gung University College of Medicine, Kaohsiung 83301, Taiwan; 2Department of Nuclear Medicine, Kaohsiung Chang Gung Memorial Hospital and Chang Gung University College of Medicine, Kaohsiung 83301, Taiwan; 3Departments of Surgery, Kaohsiung Chang Gung Memorial Hospital and Chang Gung University College of Medicine, Kaohsiung 83301, Taiwan

**Keywords:** radiofrequency ablation, hyperparathyroidism, primary hyperparathyroidism, secondary hyperparathyroidism, localization strategy, ultrasound, four-dimensional computed tomography, technetium 99m-sestamibi single-photon-emission-computed tomography/computed tomography

## Abstract

Objective: Preoperative localization in patients with primary or secondary hyperparathyroidism before radiofrequency ablation (RFA) is crucial. There is currently a lack of consensus regarding imaging protocol. Evaluating the diagnostic performance of ultrasound, four-dimensional computed tomography (4D-CT), and technetium 99m-sestamibi single-photon-emission-computed tomography/computed tomography (SPECT/CT) is necessary for RFA of hyperparathyroidism. Methods: This retrospective study recruited patients with primary or secondary hyperparathyroidism who underwent ultrasound, 4D-CT, and SPECT/CT before RFA at a single institution. The sensitivity, accuracy, and receiver operating characteristic curve analysis were used to evaluate the diagnostic performance of the imaging modalities. Results: A total of 33 patients underwent RFA for hyperparathyroidism (8 patients with primary hyperparathyroidism, 25 patients with secondary hyperparathyroidism). Ultrasound had the highest sensitivity (0.953) and accuracy (0.943), while 4D-CT had higher sensitivity and accuracy than SPECT/CT (sensitivity/accuracy, 4D-CT vs. SPECT/CT: 0.929/0.920 vs. 0.741/0.716). Combined ultrasound with 4D-CT and the three combined modalities achieved equivalent, and the highest, diagnostic performance (sensitivity 1.000, accuracy 0.989). The lesion length and volume were important predictors of the diagnostic performance of 4D-CT and SPECT/CT (area under curve of length in 4D-CT/volume in 4D-CT/length in SPECT/volume in SPECT: 0.895/0.834/0.767/0.761). Conclusion: Combined ultrasound with 4D-CT provides optimal preoperative localization prior to RFA in patients with primary or secondary hyperparathyroidism. The length and volume of parathyroid lesions are determinative of the diagnostic performance of 4D-CT and SPECT/CT.

## 1. Introduction

Hyperparathyroidism is a common endocrine disorder characterized by elevated parathyroid hormone (PTH) and serum calcium levels. This results from primary hyperparathyroidism (PHPT) associated with excessive secretion of PTH from parathyroid glands, or secondary hyperparathyroidism (SHPT), which commonly occurs in patients with end-stage renal disease [1,2].

Parathyroidectomy is the standard treatment for symptomatic hyperparathyroidism resistant to medications and asymptomatic patients who meet the criteria of surgical guidelines [2,3]. However, some patients are unsuitable candidates for surgery or decline surgery due to inability to tolerate general anesthesia, repeated neck surgery, or cosmetic concerns [4]. Ultrasound-guided minimally invasive treatments, such as radiofrequency ablation (RFA), have been suggested as safe and effective alternative treatments for hyperparathyroidism [5,6,7,8,9,10,11,12,13,14,15,16]. RFA is considered as a viable alternative treatment to surgery due to the precision, minimal invasiveness, shorter procedural time, and faster recovery time [17]. In PHPT patients, RFA can significantly reduce the parathyroid nodule volume with successful iPTH and calcium remission to normal levels [5,6,7,8,9,10,11,12]. For SHPT patients, previous studies have reported variable outcomes, with a therapeutic success rate of 37.5% after 6 months [5], and 44.1% at 1 year [15]. However, a previous study revealed that 62.5% of SHPT patients had persistent hyperparathyroidism up to 6 months after RFA [5]. The outcome of RFA in PHPT patients seems to be superior to that of SHPT patients, while the accuracy of preoperative localization may be a crucial factor in determining outcome success. The hyperplastic parathyroid gland can vary in size and typically involves all four glands, and the glands heavier than 0.5 g may be composed of nodular hyperplasia [18]. The mean weight for parathyroid adenoma is approximately 1 g [19], which is considerably larger and easier to identify than parathyroid hyperplasia. Thus, the low sensitivity for the detection of parathyroid hyperplasia may complicate preoperative localization for SHPT patients [17,20,21].

Accurate preoperative localization is an important step for RFA. The commonly applied modalities for preoperative evaluation are ultrasound and technetium 99m–sestamibi single-photon-emission-computed tomography/computed tomography (sestamibi SPECT/CT) [17,22,23,24,25,26]. One meta-analysis revealed the variability of sensitivity for each imaging modality (ultrasound 70.4–81.4%, sestamibi SPECT/CT 64–90.6%) [23]. Such data have prompted physicians to combine the different modalities to improve localization accuracy. In recent years, four-dimensional computed tomography (4D-CT) has become an increasingly popular alternative method for preoperative localization [17,20,26]. The sensitivity of 4D-CT is higher than that of scintigraphy and ultrasound, especially for multigland disease [27,28] and smaller parathyroid lesions [29], leading to speculation that 4D-CT may become the preferred first-line imaging modality.

There is currently no consensus regarding the optimal imaging protocols for localization, which depends greatly on the diagnostic performance of different imaging modalities, regional imaging capabilities, radiologist expertise, and the preference of surgeons. The present study thus aimed to establish a localization strategy prior to RFA for hyperparathyroidism.

## 2. Materials and Methods

### 2.1. Patient Selection

We herein conducted a retrospective study of patients with biochemical evidence of PHPT or SHPT who received RFA at a single institution. A total of 70 patients underwent localization strategies, including 4D-CT, sestamibi SPECT/CT, and neck ultrasound before RFA. Exclusion criteria were patients who did not undergo three image evaluations (n = 27) and those who did not undergo RFA at our institution (n = 10). A total of 33 patients were included in the analysis. Figure 1 depicts the study flowchart of the patient selection process. This study was approved by the Institutional Review Board and was conducted in accordance with the Declaration of Helsinki.

### 2.2. Image Protocol

The ultrasounds were performed with ultrasound diagnostic instruments (Siemens Acuson S3000 Ultrasound System, Siemens Medical Solutions, Illinois, USA) and line array high-frequency probes (8–10 MHz). The patient was in a supine position with mild cervical extension. 

The 4D-CT was performed with a 160-section CT scanner (Aquilion Prime SP, Canon Medical Systems Corporation, Tochigi, Japan)). The protocol consisted of non-enhanced, arterial, and delayed (venous) phase with scanning from the frontal sinus to the carina. After intravenously administering 100 mL of iopamidol in a dorsal pedal vein at a rate of 3.5 mL/s, the arterial phase was acquired using bolus tracking in the aortic arch with the threshold set at 180HU. The delayed (venous) phase was acquired 15 s after the start of the arterial phase. The patient was in a supine position, with the neck supported in an extended position and shoulders depressed. This posture can reduce beam hardening artifacts and image noise in the lower neck resulting from the clavicles and broad shoulders.

The 99 mTc sestamibi SPECT/CT imaging was performed on a Siemens Symbia T SPECT/CT gamma camera (Siemens Medical Solutions, IL, USA). After intravenously administering 1110 MBq (30 mCi) of 99 mTc sestamibi via a lower limb vein, each patient underwent early-phase planar imaging (15 min) and delayed-phase planar imaging (120 min) followed by SPECT/CT. 

### 2.3. Image Interpretation

The sestamibi SPECT/CT images were interpreted for localization before RFA by one of two nuclear medicine radiologists (P.W.W., with 42 years of experience; and Y.H.C., with 9 years of experience). The 4D-CT and ultrasound images were interpreted by one of two neuroradiologists (W.C.L., with 16 years of experience; and C.Y.L., with 1 year of experience). Positive lesions on sestamibi SPECT/CT were defined as a focal area of increased tracer uptake in the neck, mediastinum, or any potential ectopic site of the parathyroid gland showing either a progressive increase or a prolonged retention at the delayed phase. The positive lesions on 4D-CT were defined as abnormal parathyroid glands with characteristic appearance such as enhancement peaking in the arterial phase, washing out in the venous phase, low attenuation on the nonenhanced images, or additional findings with cystic change, calcification, or polar vessel sign [17,30]. The positive lesions on ultrasound were defined as abnormal parathyroid glands with characteristic appearance such as homogeneously hypoechoic to the overlying thyroid gland, larger than 1 cm in diameter, oval or multilobulated in shape, polar vessel, internal calcification, or peripheral distribution of vascularity [17,30]. Cytological examination and PTH assay were performed using ultrasound-guided fine needle aspiration to confirm parathyroid adenoma or hyperplasia for a lesion without characteristic appearance.

### 2.4. Radiofrequency Ablation Procedure and Data Collection

The RFA procedures were performed by a neuroradiologist who interpreted the 4D-CT and ultrasound images (W.C.L., with 16 years of experience). The technique of radiofrequency ablation was described in our previous study [31]. The serum biochemistry samples including calcium, phosphorus, intact parathyroid hormone, and alkaline phosphatase were obtained before RFA, one day, one week, one month, three months, six months, and one year after RFA.

### 2.5. Statistical Analysis

The standard result was defined as (1) concordant location of the positive lesion on two or three modalities; (2) the positive lesion only seen on ultrasound with characteristic appearance or with cytological confirmation; (3) the positive lesion only seen on 4D-CT with characteristic appearance. For the analysis of combined imaging modalities, if any two of the three modalities located the positive lesion within the same thyroid quadrant, it was recorded as “true positive”. The diagnostic performance of the imaging modalities to detect a parathyroid lesion was calculated using the binary classification (parathyroid adenoma or hyperplasia vs. no abnormal lesion or not parathyroid lesion). The receiver operating characteristic (ROC) curve analysis was used to determine the effect of image features on the different imaging modalities. The cut-off point of image features was determined based on the index range when Youden’s index was at maximum. Comparisons of the independent ROC curves between the diagnostic performances of the different modalities were analyzed by Hanley and McNeil [32].

## 3. Results

### 3.1. Demographic and Clinical Characteristics of Patients

In Table 1, the median age of the patients was 61 years (range 24–79 years). Of the 33 patients, 19 patients (57.6%) were women and 14 patients (42.2%) were men. There were 8 patients (24.2%) with PHPT, and 25 patients (75.8%) with SHPT. All patients with PHPT had single parathyroid adenoma and had not undergone prior parathyroidectomy. Six SHPT patients had undergone prior parathyroidectomy, five of which (83.3%) had one residual parathyroid hyperplasia, and the other (16.7%) had three residual parathyroid hyperplasia. Nineteen SHPT patients had not undergone prior parathyroidectomy, two of which (10.5%) had two parathyroid hyperplasia, three of which (15.8%) had three parathyroid hyperplasia, and the remaining fourteen patients (73.7%) had four parathyroid hyperplasia.

### 3.2. Characteristics of Positive Lesions

Table 2 presents the characteristics of the positive lesions as analyzed individually. A total of 88 positive lesions were detected on ultrasound, 4D-CT, or sestamibi SPECT/CT. In total, 85 positive lesions (96.6%) were confirmed to be parathyroid adenomas or hyperplasia; 61 of 88 (69.3%) were concordantly located on three modalities, 16 of 88 (18.2%) were concordantly located on two modalities, 6 of 88 (6.8%) were only seen on ultrasound with characteristic appearance or with cytological confirmation, and 2 of 88 (2.3%) were only seen on 4D-CT with characteristic appearance. 

Cytological examinations were performed on 17 of 88 (19.3%) positive lesions, and PTH assays were performed on 6 of 88 (6.8%) positive lesions because of the lesions without the characteristic appearance. Three positive lesions (3.4%) were confirmed not to be parathyroid tissue, one of which was thyroid Hurthle cell neoplasm in a PHPT patient, one of which was thyroid nodule in a SHPT patient, and the other was a lymph node in a SHPT patient. Of the 85 parathyroid adenomas or hyperplasia, 43 (50.6%) were from the right side and 42 (49.4%) were from the left side. Among all patients, there were no ectopic parathyroid lesions noted. 

### 3.3. Diagnostic Accuracy of the Imaging Modalities 

Table 3 illustrates the sensitivity, specificity, positive predictive value, negative predictive value, and accuracy of the different imaging modalities by lesion. Among the three modalities, ultrasound had the highest sensitivity (0.953) and accuracy (0.943), while 4D-CT exhibited a higher sensitivity and accuracy than sestamibi SPECT/CT (sensitivity/accuracy, 4D-CT vs. sestamibi SPECT/CT: 0.929/0.920 vs. 0.741/0.716). Combined ultrasound with 4D-CT, and the three combined modalities achieved the equivalent, and the highest, diagnostic performance (sensitivity 1.000, accuracy 0.989). The sensitivity and accuracy of the combination of ultrasound with sestamibi SPECT/CT were 0.976 and 0.966. Combined 4D-CT with sestamibi SPECT/CT had the lowest sensitivity (0.929) and accuracy (0.920), equivalent to 4D-CT alone.

In Table 4, we grouped the lesions according to PHPT or SHPT. Similar results for all lesions were observed in the SHPT lesions (sensitivity/accuracy; ultrasound, 4D-CT, sestamibi SPECT/CT, combined ultrasound with 4D-CT, combined ultrasound with sestamibi SPECT/CT, combined 4D-CT with sestamibi SPECT/CT, three combined modalities: 0.948/0.949, 0.922/0.924, 0.753/0.734, 1.000/1.000, 0.974/0.975, 0.922/0.924, 1.000/1.000). Combined ultrasound with 4D-CT and the three combined modalities achieved equivalent, and the highest, sensitivity (1.000) and accuracy (1.000). In the PHPT lesions, the sensitivity and accuracy of ultrasound, 4D-CT, and of the three combined modalities were equivalent (1.000 and 0.889), and higher than the sensitivity and accuracy of sestamibi SPECT/CT (0.625 and 0.556). 

### 3.4. The Effect of Image Features on Different Imaging Modalities

As shown in Table 5, we grouped the lesions according to positive or negative imaging findings. The lesion length and volume in the 4D-CT positive group was significantly longer and larger than that of the 4D-CT negative group (length, *p* = 0.01; volume, *p* = 0.07). Additionally, the lesion length and volume in the sestamibi SPECT/CT positive group was significantly longer and larger than that of the sestamibi SPECT/CT negative group (length, *p* < 0.001; volume, *p* < 0.001). There were no significant differences in lesion location, length, or volume between the ultrasound positive and negative groups.

To further verify the results, ROC analysis was used to determine the effect of image features on the different imaging modalities (Figure 2). The lesion length and volume were important predictors of the diagnostic performance of 4D-CT and sestamibi SPECT/CT (AUC of lesion length in 4D-CT 0.895, *p* = 0.001, 95% CI 0.798–0.991; AUC of lesion volume in 4D-CT 0.834, *p* = 0.007, 95% CI 0.656–1.000; AUC of lesion length in sestamibi SPECT/CT 0.767, *p* < 0.001, 95% CI 0.658–0.875; AUC of lesion volume in sestamibi SPECT/CT 0.761, *p* < 0.001, 95% CI 0.652–0.870). The diagnostic performance of ultrasound was not affected by lesion length or volume (AUC of lesion length in ultrasound 0.491, *p* = 0.950, 95% CI 0.345–0.637; AUC of lesion volume on ultrasound 0.548, *p* = 0.748, 95% CI 0.439–0.656). There was no statistical difference in the AUCs of lesion length for 4D-CT and sestamibi SPECT/CT (*p* = 0.082), nor in the AUCs of lesion volume for 4D-CT and sestamibi SPECT/CT (*p* = 0.492). 

The Youden’s index indicated that the lesion length (1.15 cm as cut-off point, sensitivity 68.4%, specificity 1.00%) and lesion volume (0.118 cm^3^ as cut-off point, sensitivity 94.9%, specificity 66.7%) were helpful to predict the image positivity on 4D-CT. The sensitivity of 4D-CT on lesions >1.15 cm was 100%, which decreased to 81.3% when the lesion length was ≤1.15 cm. The sensitivity of 4D-CT on lesions >0.118 cm^3^ was 97.4%, which decreased to 50.0% when the lesion length was ≤0.118 cm^3^. The lesion length (1.25 cm as cut-off point, sensitivity 71.4%, specificity 81.8%) and lesion volume (0.398 cm^3^ as cut-off point, sensitivity 68.3%, specificity 81.8%) were helpful to predict image positivity in sestamibi SPECT/CT. The sensitivity of sestamibi SPECT/CT on lesions >1.25 cm was 91.8%, which decreased to 50.0% when the lesion length was ≤1.25 cm. The sensitivity of sestamibi SPECT/CT on lesions >0.398 cm^3^ was 91.5%, which decreased to 52.6% when the lesion length was ≤0.398 cm^3^. 

## 4. Discussion

Our study reveals that ultrasound exhibited the highest sensitivity (0.953) and accuracy (0.943) among the imaging modalities, while 4D-CT exhibited a higher sensitivity and accuracy than sestamibi SPECT/CT (sensitivity/accuracy, 4D-CT vs. sestamibi SPECT/CT: 0.929/0.920 vs. 0.741/0.716). Combined ultrasound with 4D-CT and the three combined modalities had equivalent, and the highest, diagnostic performance (sensitivity 1.000, accuracy 0.989). Similar results were also observed in the SHPT lesions. In the PHPT lesions, the sensitivity and accuracy of ultrasound, 4D-CT, and the three combined modalities were equivalent (1.000 and 0.889), and higher than the sensitivity and accuracy of sestamibi SPECT/CT (0.625 and 0.556). The lesion length and volume were important predictors of the diagnostic performance of 4D-CT and sestamibi SPECT/CT (AUC of length in 4D-CT/volume in 4D-CT/length in sestamibi SPECT/volume in sestamibi SPECT: 0.895/0.834/0.767/0.761). With a lesion length ≤ 1.15 cm or lesion volume ≤ 0.118 cm^3^, the sensitivity of 4D-CT significantly decreased. With a lesion length ≤ 1.25 cm or lesion volume ≤ 0.398 cm^3^, the sensitivity of sestamibi SPECT/CT significantly decreased. 

The accuracy of ultrasound may be difficult to generalize as it depends on the expertise of the operator. A previous meta-analysis study revealed that the sensitivity and positive predictive values of ultrasound, sestamibi SPECT, and 4D-CT were 76.1% and 93.2%, 78.9% and 90.7%, and 89.4% and 93.5%, respectively [23]. Meanwhile, our study demonstrates that ultrasound achieved superior diagnostic performance and was not affected by lesion length or volume due to the high resolution which provides detailed images, particularly of the fusion of parathyroid lesions (Figure 3), intra-thyroidal parathyroid lesions, and small parathyroid hyperplasia attached to the multinodular thyroid (Figure 4). Additionally, a characteristic ultrasound finding, known as polar feeding vessel, significantly increases specificity for the detection of parathyroid lesions [33]. However, ultrasound has poor penetration of bone and air, which limits the detection of deeper parathyroid glands located behind the trachea, clavicle, esophagus, or in the mediastinum (Figure 5) [17]. Furthermore, the wide scanning field of 4D-CT and sestamibi SPECT/CT may provide an advantage for the detection of ectopic glands [17]. 

A previous meta-analysis reported results similar to those of our study, indicating that 4D-CT has higher diagnostic performance than sestamibi SPECT/CT (sensitivity 0.85 vs 0.68) [26]. It has additionally been noted that combined 4D-CT with sestamibi SPECT/CT does not improve diagnostic performance compared with 4D-CT alone [20]. The higher sensitivity of 4D-CT can be attributed to its high spatial resolution coupled with the characteristic enhancement features of hyperfunctioning parathyroid glands [26,30]. 4D-CT also presents advantages in patients with previous neck surgery, mild hyperparathyroidism, and normal calcemia [34,35,36]. Although, a potential note of concern with regards to 4D-CT and sestamibi SPECT is the increased risk of radiation exposure. Mahajan et al. [37] revealed that effective doses of 4D-CT and sestamibi SPECT were 10.4 and 7.8 mSv, respectively, and indicated that 4D-CT should be used judiciously in women younger than 30 years old and men younger than 20 years old due to the increased risk of thyroid cancer in individuals with radiation exposure.

A low success rate is a particular challenge for the localization of multigland PHPT and SPHT. A study by Randy Yeh et al. [20] indicated a 4D-CT sensitivity of 58%, as compared to a sestamibi SPECT/CT sensitivity of 31% in multigland disease. In our study, a total of 81 lesions were identified in 30 patients who underwent ultrasound after 4D-CT. Among these 81 lesions, the sensitivity and accuracy of ultrasound were 0.962 and 0.951, respectively. Additionally, 7 lesions were found in 3 patients who underwent ultrasound prior to 4D-CT. Among these 7 lesions, the sensitivity and accuracy of ultrasound were 0.857 and 0.857, respectively. These results indicate that using 4D-CT as reference tool may increase the diagnostic performance of ultrasound. 

In summary, the diagnostic performance of PHPT localization is high, whether using ultrasound or 4D-CT. For SHPT patients, we have developed a diagnostic algorithm for parathyroid hyperplasia localization prior to RFA (Figure 6):

(1) The imaging protocol consists of 4D-CT followed by ultrasound. RFA can be performed upon detection of four parathyroid hyperplasia. If the imaging appearances are atypical on 4D-CT or ultrasound, tissue aspiration should be considered. 

(2) Sestamibi SPECT/CT is required to determine the undetected or ectopic parathyroid hyperplasia when less than four parathyroid hyperplasia are identified on 4D-CT or ultrasound. RFA can be performed if there is no additional uptake focus.

(3) Second ultrasound with tissue aspiration may be required if additional uptake focus is detected. RFA can be performed when the lesion is confirmed to be a parathyroid hyperplasia.

(4) If additional uptake focus cannot be detected by ultrasound or is not a parathyroid hyperplasia, further observation, or appropriate treatment such as surgery may be discussed by the patient and clinical doctor.

With a suspicious lesion without characteristic image appearance, ultrasound-guided fine needle aspiration with cytological examination and PTH assay may be performed [38,39,40,41]. This procedure is a safe technique and particularly helpful in patients with previous parathyroidectomy, neck dissection, or ectopic glands. One potential complication of fine needle aspiration is tissue seeding, known as parathyromatosis, along the needle track. However, a retrospective review [42] followed 81 patients with hyperparathyroidism who underwent fine needle aspiration for a mean of 5.8 years and revealed no cases of parathyromatosis. To avoid such a complication, the operator could fix the aspiration needle in the center of the suspicious lesion and oscillate the piston with negative pressure rather than move the needle tip back and forth [38]. In addition to aspiration, shear wave elastography is another helpful non-invasive modality in diagnosing PHPT and SHPT [43,44,45]. The shear wave elastography mean value has good accuracy (92.5%/92.26%) identifying PHPT/SHPT between thyroid, and accuracy (97.5%/91.75%) identifying PHPT/SHPT between muscle [43,44]. Further, it may have the potential to distinguish parathyroid carcinoma and parathyroid benign lesion [45]. However, it has the same limitations as conventional ultrasound, such as poor penetration of bone, air, and mediastinum.

There are some limitations to the present study. First, as this was a retrospective study, the patients who did not undergo a complete image evaluation or did not subsequently undergo RFA at our institution were excluded. Second, there were no surgical reports with pathologic confirmation as the reference standard for localization. Nonetheless, all PHPT patients and 57.1% of SHPT patients achieved a successful treatment of intact parathyroid hormone at the 12-month follow-up, while the serum calcium level significantly decreased to a normal range in 85.7% of all patients. Ultimately, the choice of imaging modalities depends on several factors including availability, operator preference, and radiologist expertise. Although the ultrasound has the highest image resolution with the highest sensitivity and accuracy, sometimes the ultrasound may not be straightforward in differentiating between parathyroid lesion, lymph node, or small thyroid hypoechoic nodule.

## 5. Conclusions

Compared with 4D-CT and sestamibi SPECT/CT, ultrasound exhibited the highest image resolution with the highest sensitivity (0.953) and accuracy (0.943) for identifying PHPT and SHPT. Combined ultrasound with 4D-CT provides optimal preoperative localization prior to RFA in patients with PHPT or SHPT (sensitivity 1.000, accuracy 0.989). The length and volume of parathyroid lesions were determinative of the diagnostic performance of 4D-CT (1.15 cm and 0.118 cm^3^ as cut-off point) and sestamibi SPECT/CT (1.25 cm and 0.398 cm^3^ as cut-off point). However, the choice of imaging modalities depends on several factors including availability, operator preference, and radiologist expertise.

## Figures and Tables

**Figure 1 biomedicines-11-00672-f001:**
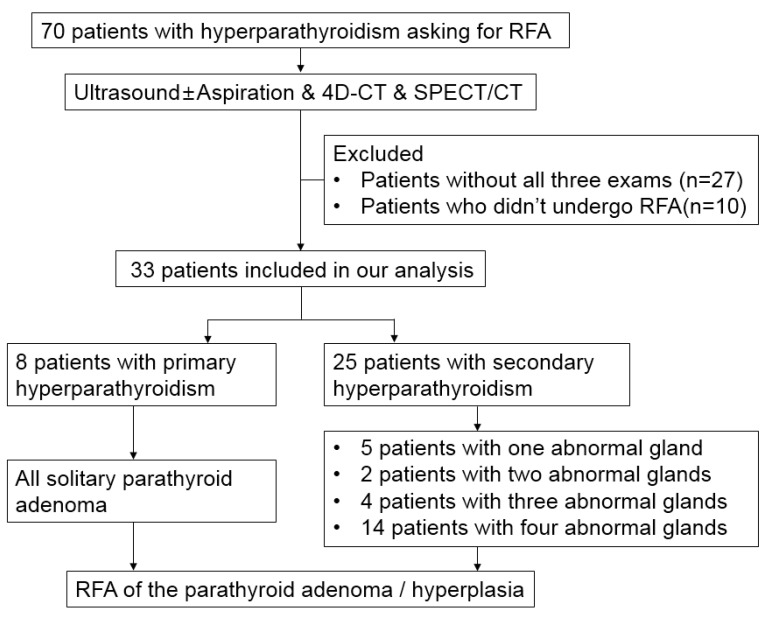
The study flowchart of patient selection process with inclusion and exclusion criteria. RFA = radiofrequency ablation. 4D-CT = four-dimensional CT. SPECT/CT = technetium 99m–sestamibi single-photon-emission-computed tomography/computed tomography.

**Figure 2 biomedicines-11-00672-f002:**
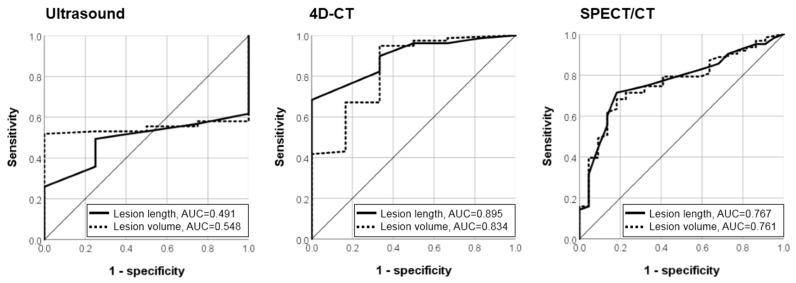
The ROC curves of lesion length and lesion volume used for predicting image positive. [Ultrasound] AUC of lesion length and lesion volume used for predicting image positive on ultrasound were 0.491 (*p* = 0.950, 95% confidence interval [CI], 0.345–0.637) and 0.548 (*p* = 0.748, 95% CI, 0.439–0.656). [4D-CT] AUC of lesion length and lesion volume used for predicting image positive on 4D-CT were 0.895 (*p* = 0.001, 95% CI, 0.798–0.991) and 0.834 (*p* = 0.007, 95% CI, 0.656–1.000). [SPECT/CT] AUC of lesion length and lesion volume used for predicting image positive on sestamibi SPECT/CT were 0.767 (*p* < 0.001, 95% CI, 0.658–0.875) and 0.761 (*p* < 0.001, 95%CI, 0.652–0.870).

**Figure 3 biomedicines-11-00672-f003:**
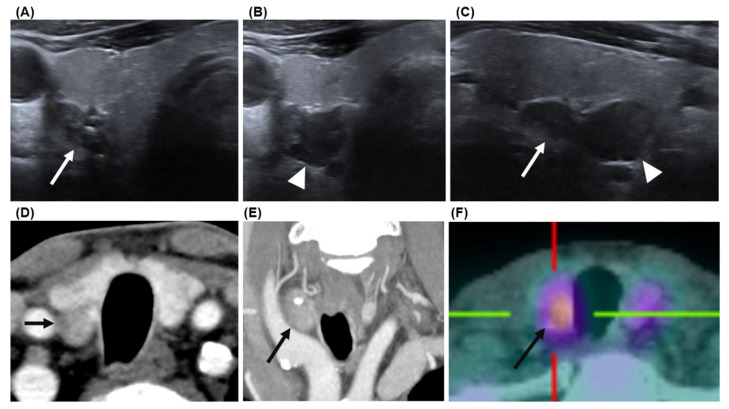
A 63-year-old man with secondary hyperparathyroidism. Ultrasound revealed (**A**) right superior parathyroid hyperplasia with internal calcification (white arrow) in axial view, (**B**) right inferior parathyroid hyperplasia (white arrowhead) in axial view, (**C**) right superior parathyroid hyperplasia (white arrow), and right inferior parathyroid hyperplasia (white arrowhead) in sagittal view. 4D-CT revealed fusion of right superior and inferior parathyroid hyperplasia (black arrow) in axial view (**D**) and in coronal view (**E**). Sestamibi SPECT/CT revealed fusion of right superior and inferior parathyroid hyperplasia (black arrow) in axial view (**F**).

**Figure 4 biomedicines-11-00672-f004:**
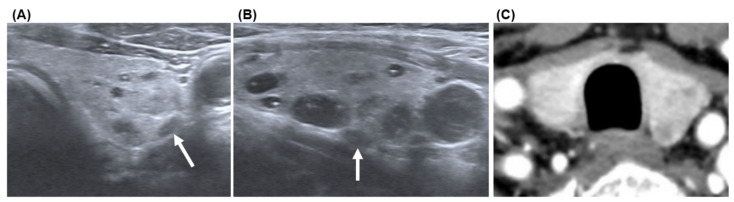
A 60-year-old man with secondary hyperparathyroidism. Ultrasound revealed left superior parathyroid hyperplasia (white arrow) in axial view (**A**) and in sagittal view (**B**). The left superior parathyroid hyperplasia was not seen on 4D-CT (**C**) because the parathyroid hyperplasia is small and attached to the multinodular thyroid.

**Figure 5 biomedicines-11-00672-f005:**
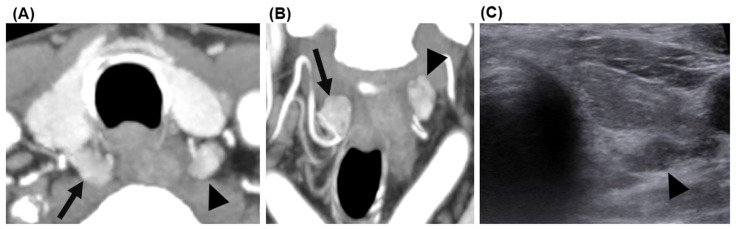
A 63-year-old man with secondary hyperparathyroidism. 4D-CT revealed right superior parathyroid hyperplasia (arrow) and left superior parathyroid hyperplasia (arrowhead) in axial view (**A**) and in coronal view (**B**). Ultrasound revealed left superior parathyroid hyperplasia (arrowhead) in axial view (**C**). The right superior parathyroid hyperplasia was not seen because it was behind the trachea.

**Figure 6 biomedicines-11-00672-f006:**
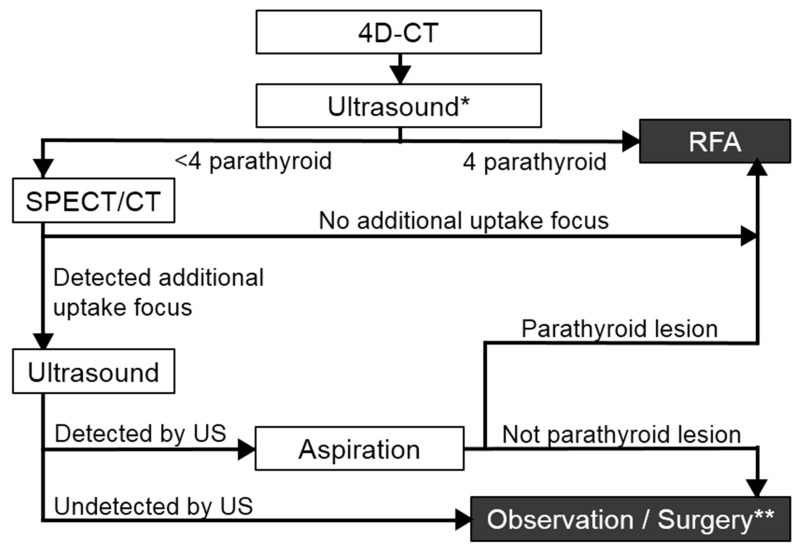
The diagnostic algorithm of localization before radiofrequency ablation for secondary hyperparathyroidism. * Ultrasound-guided tissue aspiration in suspicious lesions with atypical features on 4D-CT or ultrasound. ** Dependent on decision of the patient and clinical doctor.

**Table 1 biomedicines-11-00672-t001:** Demographic and Clinical Characteristics of Patients.

	Primary Hyperparathyroidism (n = 8)		Secondary Hyperparathyroidism (n = 25)		All (n = 33)
Age (year)	52.5 (47.75–61.25)		63 (49–68)		61 (49–67)
Gender (Male)	3 (37.5%)		11 (44.0%)		14 (42.4%)
Serum biochemistry before RFA					
I-PTH (pg/mL)	177.0 (145.9–232.1)		1356.9 (1038.3–1979.8)		1259.3 (643.5–1829.3)
Calcium (mg/dL)	11.4 (11.3–11.6)		10.4 (9.7–11.0)		10.6 (9.9–11.3)
Phosphorus (mg/dL)	2.6 (2.4–3.1)		5.3 (4.0–6.1)		4.7 (3.3–6.1)
ALK-P (U/L)	90.5 (86.0–175.3)		135.5 (88.0–210.75)		115.5(86.0–195.0)
Patient-based analysis (number of patients)					
Previous parathyroidectomy	No (n = 8) (100%)	Yes (n = 0) (0.0%)	No (n = 19) (76.0%)	Yes (n = 6) (24.0%)	
One gland	8 (100%)	0 (0.0%)	0 (0.0%)	5 (83.3%)	13 (39.4%)
Two glands	0 (0.0%)	0 (0.0%)	2 (10.5%)	0 (0.0%)	2 (6.1%)
Three glands	0 (0.0%)	0 (0.0%)	3 (15.8%)	1 (16.7%)	4 (12.1%)
Four glands	0 (0.0%)	0 (0.0%)	14 (73.7%)	0 (0.0%)	14 (42.4%)

Non-normally distributed continuous variables were presented as median (interquartile range). Categorical variables were presented as n (%). Abbreviations: I-PTH = intact parathyroid hormone, ALK-P = alkaline phosphatase.

**Table 2 biomedicines-11-00672-t002:** Characteristics of Positive Lesions (Number of Lesions).

	Primary Hyperparathyroidism (n = 9)	Secondary Hyperparathyroidism (n = 79)	Total (n = 88)
Concordant location in three modalities	5 (55.6%)	56 (70.9%)	61 (69.3%)
Concordant location in two modalities	3 (33.3%)	13 (16.5%)	16 (18.2%)
Only seen in ultrasound *	0 (0.0%)	6 (7.6%)	6 (6.8%)
Only seen in 4D-CT	0 (0.0%)	2 (2.5%)	2 (2.3%)
Not parathyroid gland	1 (thyroid Hurthle cell neoplasm) (11.1%)	2 (thyroid nodule and lymph node) (2.5%)	3 (3.4%)
Total number of parathyroid adenoma/hyperplasia (n= 85)			
Right superior	0 (0.00%)	16 (20.8%)	16 (18.8%)
Right inferior	4 (50.0%)	23 (29.9%)	27 (31.8%)
Left superior	3 (37.5%)	17 (22.1%)	20 (23.5%)
Left inferior	1 (12.5%)	21 (27.3%)	22 (25.9%)

Categorical variables were presented as n (%). * With characteristic appearance or with cytological confirmation.

**Table 3 biomedicines-11-00672-t003:** Diagnostic Accuracy of Imaging Modalities Analyzed by Lesion (All Lesions).

%	US	4D-CT	SPECT	US + 4D-CT	US + SPECT	4D-CT + SPECT	All
Sensitivity	0.953 (81/85)	0.929 (79/85)	0.741 (63/85)	1.000 (85/85)	0.976 (83/85)	0.929 (79/85)	1.000 (85/85)
Specificity	0.667 (2/3)	0.667 (2/3)	0.000 (0/3)	0.667 (2/3)	0.667 (2/3)	0.667 (2/3)	0.667 (2/3)
PPV	0.988 (81/82)	0.988 (79/80)	0.955 (63/66)	0.988 (85/86)	0.988 (83/84)	0.988 (79/80)	0.988 (85/86)
NPV	0.333 (2/6)	0.250 (2/8)	0.000 (0/22)	1.000 (2/2)	0.500 (2/4)	0.250 (2/8)	1.000 (2/2)
Accuracy	0.943 (83/88)	0.920 (81/88)	0.716 (63/88)	0.989 (87/88)	0.966 (85/88)	0.920 (81/88)	0.989 (87/88)

Abbreviations: PPV = positive predictive value, NPV = negative predictive value, US = ultrasound.

**Table 4 biomedicines-11-00672-t004:** Diagnostic Accuracy of Imaging Modalities Analyzed by Lesion (Primary or Secondary Hyperparathyroidism).

%	US	4D-CT	SPECT	US + 4D-CT	US + SPECT	4D-CT + SPECT	All
Primary hyperparathyroidism (n = 9)							
Sensitivity	1.000 (8/8)	1.000 (8/8)	0.625 (5/8)	1.000 (8/8)	1.000 (8/8)	1.000 (8/8)	1.000 (8/8)
Specificity	0.000 (0/1)	0.000 (0/1)	0.000 (0/1)	0.000 (0/1)	0.000 (0/1)	0.000 (0/1)	0.000 (0/1)
PPV	0.889 (8/9)	0.889 (8/9)	0.833 (5/6)	0.889 (8/9)	0.889 (8/9)	0.889 (8/9)	0.889 (8/9)
NPV	0.000 (0/0)	0.000 (0/0)	0.000 (0/3)	0.000 (0/0)	0.000 (0/0)	0.000 (0/0)	0.000 (0/0)
Accuracy	0.889 (8/9)	0.889 (8/9)	0.556 (5/9)	0.889 (8/9)	0.889 (8/9)	0.889 (8/9)	0.889 (8/9)
Secondary hyperparathyroidism (n = 79)							
Sensitivity	0.948 (73/77)	0.922 (71/77)	0.753 (58/77)	1.000 (77/77)	0.974 (75/77)	0.922 (71/77)	1.000 (77/77)
Specificity	1.000 (2/2)	1.000 (2/2)	0.000 (0/2)	1.000 (2/2)	1.000 (2/2)	1.000 (2/2)	1.000 (2/2)
PPV	1.000 (73/73)	1.000 (71/71)	0.967 (58/60)	1.000 (77/77)	1.000 (75/75)	1.000 (71/71)	1.000 (77/77)
NPV	0.333 (2/6)	0.250 (2/8)	0.000 (0/19)	1.000 (2/2)	0.500 (2/4)	0.250 (2/6)	1.000 (2/2)
Accuracy	0.949 (75/79)	0.924 (73/79)	0.734 (58/79)	1.000 (79/79)	0.975 (77/79)	0.924 (73/79)	1.000 (79/79)

Abbreviations: PPV = positive predictive value, NPV = negative predictive value, US = ultrasound.

**Table 5 biomedicines-11-00672-t005:** The Effect of Image Features on Different Imaging Modalities.

All Patients											
		US(+) (n = 81)	US(-) (n = 4)	*p*		4D-CT(+) (n = 79)	4D-CT(-) (n = 6)	*p*	SPECT(+) (n = 63)	SPECT(-) (n = 22)	*p*
Lesion location (n)	RS	14	2	0.165		13	3	0.077	9	7	0.343
	RI	25	2			27	0		21	6	
	LS	20	0			18	2		15	5	
	LI	22	0			21	1		18	4	
Lesion length (cm)		1.30 (1.10–1.75)	1.30 (1.23–1.53)	0.960		1.40 (1.10–1.80)	0.78 (0.68–1.10)	0.01 *	1.50 (1.10–2.00)	1.10 (0.80–1.20)	<0.001 *
Lesion volume (cm^3^)		0.48 (0.21–1.04)	0.41 (0.37–0.43)	0.756		0.48 (0.24–1.08)	0.09 (0.05–0.39)	0.07 *	0.55 (0.31–1.40)	0.25 (0.14–0.37)	<0.001 *
Secondary Hyperparathyroidism											
		US(+) (n = 73)	US(-) (n = 4)	*p*	4D-CT(+) (n = 71)		4D-CT(-) (n = 6)	*p*	SPECT(+) (n = 58)	SPECT(-) (n = 22)	*p*
Location (n)	RS	14	2	0.219	13		3	0.095	9	7	0.289
	RI	21	2		23		0		20	3	
	LS	17	0		15		2		12	5	
	LI	21	0		20		1		17	4	
Lesion length (cm)		1.30 (1.10–1.65)	1.30 (1.23–1.53)	0.886	1.40 (1.10–1.70)		0.78 (0.68–1.10)	0.001 *	1.50 (1.10–1.83)	1.10 (0.80–1.20)	<0.001 *
Lesion volume (cm^3^)		0.48 (0.21–1.04)	0.41 (0.37–0.43)	0.799	0.48 (0.24–1.08)		0.09 (0.05–0.39)	0.008 *	0.53 (0.30–1.23)	0.24 (0.13–0.36)	0.001 *

Abbreviations: I-PTH = intact parathyroid hormone, ALK-P = alkaline phosphatase, US = ultrasound, (+) = positive, (-) = negative, RS = right superior, RI = right lower, LS = left superior, LI = left inferior. p Mann—Whitney U test. Non-normally distributed continuous variables were presented as median (interquartile range). Categorical variables were presented as n. *p* < 0.01 was marked as *.

## Data Availability

Data available on request due to restrictions (e.g., privacy or ethical)**.** The data presented in this study are available on request from the corresponding author. The data are not publicly available due to privacy of Kaohsiung Chang Gung Memorial Hospital and Chang Gung University College of Medicine.

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
