# Peer review of "Localization Strategy Prior to Radiofrequency Ablation for Primary and Secondary Hyperparathyroidism"

_biomedicines, 2023, doi:10.3390/biomedicines11030672_

Round 1
Reviewer 1 Report
In the manuscript " Localization Strategy prior to Radiofrequency Ablation for Primary and Secondary Hyperparathyroidism", the authors present a very interesting retrospective study on the localization strategy prior to radioablation of the parathyroid gland. The paper is well written, and the facts are well presented, however, minor criticisms are present, as follows:
- I consider that it should be clearly stated from the abstract that a cytopathological evaluation prior to RFA was done, also excluding malignancy. This fact is not stated in the paper. Please correct.
- You did not discuss the importance of elastography, especially shear wave elastography in the localization of both primary and secondary hyperparathyroidism. This subject is well established in the literature.
- Please detail how do you perform the RFA procedure.
- Please rewrite the conclusion to better state the importance of the subject.

Author Response
- I consider that it should be clearly stated from the abstract that a cytopathological evaluation prior to RFA was done, also excluding malignancy. This fact is not stated in the paper. Please correct.
Cytological examinations were performed on 17 of 88 (19.3%) positive lesions, and PTH assays were performed on 6 of 88 (6.8%) positive lesions because of the lesions without the characteristic appearance. Three positive lesions (3.4%) were confirmed to be not parathyroid tissue, one of which was thyroid Hurthle cell neoplasm in a PHPT patient, one of which was thyroid nodule in a SHPT patient, and the other was a lymph node in a SHPT patient. (line 205-209)
- You did not discuss the importance of elastography, especially shear wave elastography in the localization of both primary and secondary hyperparathyroidism. This subject is well established in the literature.
In addition to aspiration, the shear wave elastography is another helpful non-invasive modality in diagnosing PHPT and SHPT. [43-45] The shear wave elastography-mean value has well accuracy (92.5%/92.26%) to identify PHPT/SHPT between thyroid, and ac-curacy (97.5%/91.75%) to identify PHPT/SHPT between muscle. [43,44] Further, it may have the potential to distinguish parathyroid carcinoma and parathyroid benign lesion. [45] However, it has the same limitations as conventional ultrasound, such as poor pene-tration of bone, air and mediastinum. (line400-407)
- Please detail how do you perform the RFA procedure.
The technique of radiofrequency ablation was descripted in our previous study. (line159-160)
- Please rewrite the conclusion to better state the importance of the subject.
Revise as suggested.
Reviewer 2 Report
The study evaluated the diagnostic performance of ultrasound, four-dimensional computed tomography (4D-CT), and technetium 99m-sestamibi single-photon-emission-computed tomography/computed tomography (SPECT/CT) for confirming localization strategy prior to radiofrequency ablation for primary and secondary hyperparathyroidism. The authors found that combined ultrasound with 4D-CT provides optimal preoperative localization prior to RFA in patients with primary or secondary hyperparathyroidism. The length and volume of parathyroid lesions are determinative of the diagnostic performance of 4D-CT and SPECT/CT.
In general, the topic of the study is novel. But the patient representation was low. More patient should be included.
Detailed evaluation of specific deficiencies with suggestions for improvements:
1, Line 31. The full name of AUC should be given for appearing for the first time?
2, Line 119. “Line array high-frequency probes.” Please give the specific probe frequency.
3, Did every patient have cytological examination and PTH determination? What is the “golden standard”?
Author Response
- Line 31. The full name of AUC should be given for appearing for the first time?
Revise as suggested.
- Line 119. “Line array high-frequency probes.” Please give the specific probe frequency.
Revise as suggested.
- Did every patient have cytological examination and PTH determination? What is the “golden standard”?
Cytological examinations were performed on 17 of 88 (19.3%) positive lesions, and PTH assays were performed on 6 of 88 (6.8%) positive lesions because of the lesions without the characteristic appearance. (line 205-207) The standard result was described in the statistical analysis. (line 166-171)
Round 2
Reviewer 2 Report
There are no more questions.